# 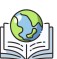 GeoLM: Empowering Language Models for Geospatially Grounded Language Understanding

**Zekun Li**ᴹ   **Wenxuan Zhou**ᕗ   **Yao-Yi Chiang**ᴹ   **Muhao Chen**ᕗᴬ

ᴹDepartment of Computer Science and Engineering, University of Minnesota, Twin Cities

ᕗDepartment of Computer Science, University of Southern California

ᴬDepartment of Computer Science, University of California, Davis

{li002666,yaoyi}@umn.edu; zhouwenx@usc.edu; muhchen@ucdavis.edu

## Abstract

Humans subconsciously engage in geospatial reasoning when reading articles. We recognize place names and their spatial relations in text and mentally associate them with their physical locations on Earth. Although pretrained language models can mimic this cognitive process using linguistic context, they do not utilize valuable geospatial information in large, widely available geographical databases, e.g., OpenStreetMap. This paper introduces GEOLM (🌐📖), a geospatially grounded language model that enhances the understanding of geo-entities in natural language. GEOLM leverages geo-entity mentions as anchors to connect linguistic information in text corpora with geospatial information extracted from geographical databases. GEOLM connects the two types of context through contrastive learning and masked language modeling. It also incorporates a spatial coordinate embedding mechanism to encode distance and direction relations to capture geospatial context. In the experiment, we demonstrate that GEOLM exhibits promising capabilities in supporting toponym recognition, toponym linking, relation extraction, and geo-entity typing, which bridge the gap between natural language processing and geospatial sciences. The code is publicly available at https://github.com/knowledge-computing/geolm.

## 1 Introduction

Spatial reasoning and semantic understanding of natural language text arise from human communications. For example, "I visited Paris in Arkansas to see the smaller Eiffel Tower", a human can easily recognize the *toponyms* "Eiffel Tower", "Paris", "Arkansas", and the spatial relation "in". Implicitly, a human can also infer that this "Eiffel Tower" might be a replica[1] of the original one in Paris,

---

[1] The small Eiffel Tower in Paris, Arkansas, United States: https://www.arkansas.com/paris/accommodations/eiffel-tower-park

France. Concretely, the process of geospatially grounded language understanding involves core tasks such as recognizing geospatial concepts being described, inferring the identities of those concepts, and reasoning about their spatially qualified relations. These tasks are essential for applications that involve the use of place names and their spatial relations, such as social media message analysis (Hu et al., 2022), emergency response (Gritta et al., 2018b), natural disaster analysis (Wang et al., 2020), and geographic information retrieval (Wallgrün et al., 2018).

Pretrained language models (PLMs; Devlin et al. 2019; Liu et al. 2019; Raffel et al. 2020) have seen broad adaptation across various domains such as biology (Lee et al., 2020), healthcare (Alsentzer et al., 2019), law (Chalkidis et al., 2020; Douka et al., 2021), software engineering (Tabassum et al., 2020), and social media (Röttger and Pierrehumbert, 2021; Guo et al., 2021). These models benefit from in-domain corpora (e.g., PubMed for the biomedical domain) to learn domain-specific terms and concepts. Similarly, a geospatially grounded language model requires training with geo-corpora. The geo-corpora should cover worldwide geo-entity names and their variations, geographical locations, and spatial relations to other geo-entities. Although some geo-corpora are available, e.g., LGL (Lieberman et al., 2010) and SpatialML (Mani et al., 2010), the sizes of the datasets are relatively small.

In contrast, Wikipedia stores many articles describing places worldwide and can serve as comprehensive geo-corpora for geospatially grounded language understanding. However, training with Wikipedia only solves partial challenges in geospatial grounding, as it only provides the linguistic context of a geo-entity with sentences describing history, demographics, climate, etc. The information about the geospatial neighbors of a geo-entity is still missing. On the other hand, large-

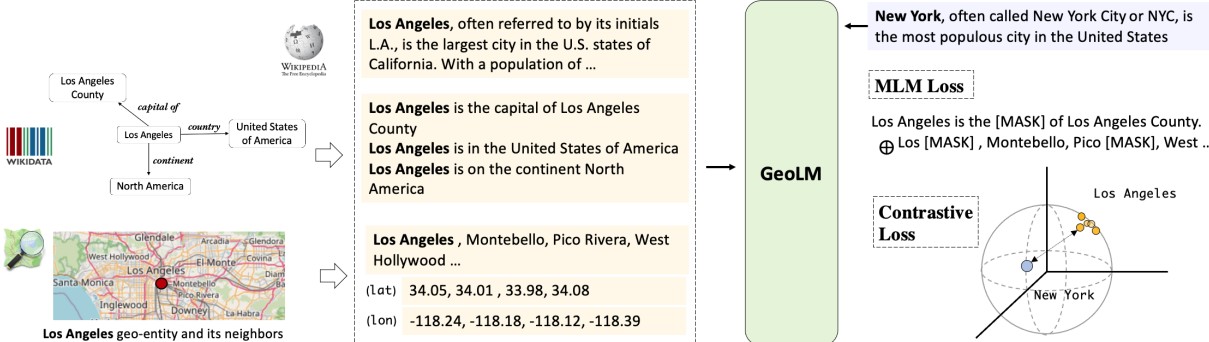

Figure 1: Outline of GEOLM. Wikipedia and Wikidata form the NL corpora, and OpenStreetMap (OSM) form the pseudo-sentence corpora (Details in §2.2). GEOLM takes the NL and pseudo-sentence corpora as input, then pretrain with MLM and contrastive loss using geo-entities as anchors. (See §2.3)

scale geographical databases (e.g., OpenStreetMap) and knowledge bases (e.g., Wikidata) can provide extensive amounts of geo-entity locations and geospatial relations, enriching the information sourced from Wikipedia. Additionally, the geo-locations from OpenStreetMap can help connecting the learned geo-entity representations to physical locations on the Earth.

To address these challenges, we propose GE-OLM, a language model specifically designed to support geospatially grounded natural language understanding. Our model aims to enhance the comprehension of places, types, and spatial relations in natural language. We use Wikipedia, Wikidata, and OSM together to create geospatially grounded training samples using geo-entity names as anchors. GEOLM verbalizes the Wikidata relations into natural language and aligns the linguistic context in Wikipedia and Wikidata with the geospatial context in OSM. Since existing language models do not take geocoordinates as input, GEOLM further incorporates a spatial coordinate embedding module to learn the distance and directional relations between geo-entities. During inference, our model can take either natural language or geographical subregion (i.e., a set of nearby geo-entities) as input and make inferences relying on the aligned linguistic-geospatial information. We employ two training strategies: contrastive learning (Oord et al., 2018) between natural language corpus and linearized geographic data, and masked language modeling (Devlin et al., 2019) with a concatenation of these two modalities. We treat GEOLM as a versatile model capable of addressing various geographically related language understanding tasks, such as toponym recognition, toponym linking, and geospatial relation extraction.

## 2 GEOLM

This section introduces GEOLM's mechanism for representing both the linguistic and geospatial context (§2.1), followed by the detailed development process of pretraining tasks (§2.3) and corpora (§2.2), and how GEOLM is further adapted to various downstream geospatial NLU tasks (§2.4).

### 2.1 Representing Linguistic and Geospatial Context

The training process of GEOLM aims to simultaneously learn the linguistic and geospatial context, aligning them in the same embedding space to obtain geospatially grounded language representations. The ***linguistic context*** refers to the sentential context of the geographical entity (i.e., geo-entity). The linguistic context contains essential information to specify a geo-entity, including sentences describing geography, environment, culture, and history. Additionally, geo-entities exhibit strong correlations with neighboring geo-entities (Li et al., 2022b). We refer to these neighboring geo-entities as the ***geospatial context***. The geospatial context encompasses locations and implicit geo-relations of the neighbors to the center geo-entity. [2]

To capture the linguistic context, GEOLM takes natural sentences as input and generates entity-level representations by averaging the token representations within the geo-entity name span. For the geospatial context, GEOLM follows the geospatial context linearization method and the spatial embedding module in our previous work SPABERT (Li et al., 2022b), a PLM that generates geospatially contextualized entity representations using

---

[2]Here, we assume that all geo-entities in the geographical dataset are represented as points.

**NL Input**

| Tokens | [CLS] | Los | Angeles | is | the | commercial | , | financial | and | cultural | [SEP] |
|---|---|---|---|---|---|---|---|---|---|---|---|
| Position ID | 0 | 1 | 2 | 3 | 4 | 5 | 6 | 7 | 8 | 9 | 10 |
| Segment ID | 0 | 0 | 0 | 0 | 0 | 0 | 0 | 0 | 0 | 0 | 0 |
| X-Coord | DSEP | DSEP | DSEP | DSEP | DSEP | DSEP | DSEP | DSEP | DSEP | DSEP | DSEP |
| Y-Coord | DSEP | DSEP | DSEP | DSEP | DSEP | DSEP | DSEP | DSEP | DSEP | DSEP | DSEP |

$\oplus$ *concat*

**Geospatial Input**

| Tokens | Los | Angeles | [SEP] | Glen | ##dale | [SEP] | Pasadena | [SEP] | Al | ##ham |
|---|---|---|---|---|---|---|---|---|---|---|
| Position ID | 0 | 1 | 2 | 3 | 4 | 5 | 6 | 7 | 8 | 9 |
| Segment ID | 1 | 1 | 1 | 1 | 1 | 1 | 1 | 1 | 1 | 1 |
| X-Coord | 34.05 | 34.05 | DSEP | 34.17 | 34.17 | DSEP | 34.16 | DSEP | 34.08 | 34.08 |
| Y-Coord | -118.24 | -118.24 | DSEP | -118.25 | -118.25 | DSEP | -118.13 | DSEP | -118.13 | -118.13 |

Figure 2: Sample inputs to GEOLM. Note that segment IDs for the NL tokens are zeros, and for the pseudo-sentence tokens are ones.

point geographic data. Given a center geo-entity and its spatial neighbors, GEOLM linearizes the geospatial context by sorting the neighbors in ascending order based on their geospatial distances from the center geo-entity. GEOLM then concatenates the name of the center geo-entity and the sorted neighbors to form a *pseudo-sentence*. To preserve directional relations and relative distances, GEOLM employs the geocoordinates embedding module, which takes the geocoordinates as input and encodes them with a sinusoidal position embedding layer.

To enable GEOLM to process both natural language text and geographical data, we use the following types of position embedding mechanisms for each token and the token embedding (See Fig. 2). By incorporating these position embedding mechanisms, GEOLM can effectively process both natural language text and geographical data, allowing the model to capture and leverage spatial information.

***Position ID*** describes the index position of the token in the sentence. Note that the position ID for both the NL and geospatial input starts from zero.

***Segment ID*** indicates the source of the input tokens. If the tokens belong to the natural language input, then the segment ID is zero; otherwise one.

***X-coord*** and ***Y-coord*** are inputs for the spatial coordinate embedding. Tokens within the same geo-entity name span share the same *X-coord* and *Y-coord* values. Since NL tokens do not have associated geocoordinate information, we set their *X-coord* and *Y-coord* to be DSEP, which is a constant value as distance filler.

In addition, GEOLM projects the geocoordinates

(*lat, lng*) into a 2-dimensional World Equidistant Cylindrical coordinate system EPSG:4087.[3] This is because (*lat, lng*) represent angle-based values that model the 3D sphere, whereas coordinates in EPSG:4087 are expressed in Cartesian form. Additionally, when generating the pseudo-sentence, we sort neighbors of the center geo-entity based on the Haversine distance which reflects the geodesic distance on Earth instead of the Euclidean distance.

## 2.2 Pretraining Corpora

We divide the training corpora into two parts: 1) pseudo-sentence corpora from a geographical dataset, OpenStreetMap (OSM), to provide the geospatial context; 2) natural language corpora from Wikipedia and verbalized Wikidata to provide the linguistic context.

**Geographical Dataset.** OpenStreetMap (OSM) is a crowd-sourced geographical database containing a massive amount of point geo-entities worldwide. In addition, OSM stores the correspondence from OSM geo-entity to Wikipedia and Wikidata links. We preprocess worldwide OSM data and gather the geo-entities with Wikidata and Wikipedia links to prepare *paired* training data used in contrastive pretraining. To linearize geospatial context and prepare the geospatial input in Fig. 2, For each geo-entity, we retrieve its geospatial neighbors and construct pseudo-sentences (See Fig. 1) by concatenating the neighboring geo-entity names after sorting the neighbors by distance. In the end, we generate 693,308 geo-entities with the same number of pseudo-sentences in total.

---

[3] EPSG:4087: https://epsg.io/4087

**Natural Language Text Corpora.** We prepare the text corpora from Wikipedia and Wikidata. Wikipedia provides a large corpus of encyclopedic articles, with a subset describing geo-entities. We first find the articles describing geo-entities by scraping all the Wikipedia links pointed from the OSM annotations, then break the articles into sentences and adopt Trie-based phrase matching (Hsu and Ottaviano, 2013) to find the sentences containing the name of the corresponding OSM geo-entity. The training samples are paragraphs containing *at least* one corresponding OSM geo-entity name. For Wikidata, the procedure is similar to Wikipedia. We collect the Wikidata geo-entities using the QID identifier pointed from the OSM geo-entities. Since Wikidata stores the relations as triples, we convert the relation triples to natural sentences with a set of pre-defined templates (See example in Fig. 1). After merging the Wikipedia and Wikidata samples, we gather 1,458,150 sentences/paragraphs describing 472,067 geo-entities.

## 2.3 Pretraining Tasks

We employ two pretraining tasks to establish connections between text and geospatial data, enabling GEOLM to learn geospatially grounded representations of natural language text.

The first is a **contrastive learning** task using an InfoNCE loss (Oord et al., 2018), which contrasts between the geo-entity features extracted from the *two* modalities. This loss encourages GEOLM to generate similar representations for the same geo-entity, regardless of whether the representation is contextualized based on the linguistic or geospatial context. Simultaneously, GEOLM learns to distinguish between geo-entities that share the same name by maximizing the distances of their representations in the embedding space.

Formally, let the training data $\mathcal{D}$ consist of pairs of samples $(s_i^{nl}, s_i^{geo})$, where $s_i^{nl}$ is a linguistic sample (a natural sentence or a verbalized relation from a knowledge base), and $s_i^{geo}$ is a pseudo-sentence created from the geographic data. Both samples mention the same geo-entity. Let $f(\cdot)$ be GEOLM that takes both $s_i^{nl}$ and $s_i^{geo}$ as input and produces entity-level representation $\mathbf{h}_i^{nl} = f(s_i^{nl})$ and $\mathbf{h}_i^{geo} = f(s_i^{geo})$. Then the loss function is:

$$\mathcal{L}_i^{contrast} = -\log \frac{e^{\mathrm{sim}(\mathbf{h}_i^{nl}, \mathbf{h}_i^{geo})/\tau}}{\sum_{j=1}^{2N} \mathbb{1}_{[j \neq i]} e^{\mathrm{sim}(\mathbf{h}_i^{nl}, \mathbf{h}_j^{geo})/\tau}},$$

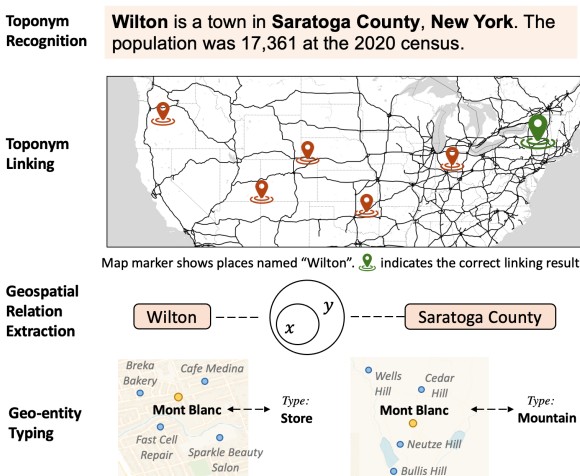

Figure 3: Downstream tasks to evaluate geospatially grounded language understanding.

where $\tau$ is a temperature, and $\mathrm{sim}(\cdot)$ denotes the cosine similarity.

To improve GEOLM's ability to disambiguate geo-entities, we include in-batch hard negatives comprising geo-entities with identical names. As a result, each batch is composed of 50% random negatives and 50% hard negatives.

Additionally, we employ a **masked language modeling** task (Devlin et al., 2019) on a concatenation of the paired natural language sentence and geographical pseudo-sentence (Fig. 2). This task encourages GEOLM to recover masked tokens by leveraging both linguistic and geographical data.

## 2.4 Downstream Tasks

Our study further adapts GEOLM to several downstream tasks to demonstrate its ability for geospatially grounded language understanding, including toponym recognition, toponym linking, geo-entity typing, and geospatial relation extraction (Fig. 3).

**Toponym recognition** or geo-tagging (Gritta et al., 2020) is to extract toponyms (i.e., place names) from unstructured text. This is a crucial step in recognizing geo-entities mentioned in the text before inferring their identities and relationships in subsequent tasks. We frame this task as a multi-class sequence tagging problem, where tokens are classified into one of three classes: B-topo (beginning of a toponym), I-topo (inside a toponym), or O (non-toponym). To accomplish this, we append a fully connected layer to GEOLM and train the model end-to-end on a downstream dataset to classify each token from the text input.

**Toponym linking**, also referred to as toponym res-

olution and geoparsing (Gritta et al., 2020, 2018a; Hu et al., 2022), aims to infer the identity or geoco-ordinates of a mentioned geo-entity in the text by grounding geo-entity mention to the correct record in a geographical database, which might contain many candidate entities with the same name as the extracted toponym from the text. During inference, we process the candidate geo-entities from the ge-ographical databases the same way as during pre-training, where nearby neighbors are concatenated together to form pseudo-sentences. For this task, we perform zero-shot linking by directly applying GEOLM without fine-tuning. GEOLM extracts representations of geo-entities from linguistic data and calculates representations for all candidate en-tities. After obtaining the representations for both query and candidate geo-entities, the linking results are formed as a ranked list of candidates sorted by cosine similarity in the feature space.

**Geo-entity typing** is the task of categorizing the types of locations in a geographical database (e.g., OpenStreetMap) (Li et al., 2022b). Geo-entity typ-ing helps us understand the characteristics of a region and can be useful for location-based recom-mendations. We treat this task as a classification problem and append a one-layer classification head after GEOLM. We train GEOLM to predict the type of the central geo-entity given a subregion of a geo-graphical area. To accomplish this, we construct a pseudo-sentence following Fig. 1, then compute the representation of the geo-entity using GEOLM and feed the representation to the classification head for training.

**Geospatial relation extraction** is the task of clas-sifying topological relations between a pair of lo-cations (Mani et al., 2010). We treat this task as an entity pair classification problem. We compute the average embedding of tokens to form the entity embedding and then concatenate the embeddings of the subject and object entities to use as input for the classifier, then predict the relationship type with a softmax classifier.

## 3 Experiments

We hereby evaluate GEOLM on the aforemen-tioned four downstream tasks. All compared mod-els (except GPT3.5) are finetuned on task-specific datasets for toponym recognition, geo-entity typ-ing, and geospatial relation extraction.[4]

---

[4]Toponym linking is an unsupervised task.

### 3.1 Toponym Recognition

**Task Setup.** We adopt GEOLM for toponym recog-nition on the GeoWebNews (Gritta et al., 2020) dataset. This dataset contains 200 news articles with 2,601 toponyms.[5] The annotation includes the start and end character positions (i.e., the spans) of the toponyms in paragraphs. We use 80% for training and 20% for testing.

**Evaluation Metrics.** We report precision, recall, and F1, and include both token-level scores and entity-level scores. For entity-level scores, a pre-diction is considered correct only when it matches exactly with the ground-truth mention (i.e., no miss-ing or partial overlap).

**Models in Comparison.** We compare GEOLM with fine-tuned PLMs including BERT, SimCSE-BERT[6] (Gao et al., 2021) , SpanBERT (Joshi et al., 2020) and SapBERT (Liu et al., 2021). SpanBERT is a BERT-based model with span prediction pre-training. Instead of masking out random tokens, SpanBERT masks out a continuous span and tries to recover the missing parts using the tokens right next to the masking boundary. SapBERT learns to align biomedical entities through self-supervised contrastive learning.[7] We compare all models in base versions.

**Results and Discussion.** Tab. 1 shows that our GEOLM yields the highest entity-level F1 score, with BERT being the close second. Since GE-OLM's weights are initialized from BERT, the improvement over BERT shows the effectiveness of in-domain training with geospatial grounding. For token-level results, SpanBERT has the best F1 score for I-topo, showing that span prediction during pretraining is beneficial for predicting the continuation of toponyms. However, GEOLM is better at predicting the start token of the toponym, which SpanBERT does not perform as well.

### 3.2 Toponym Linking

**Task Setup.** We run unsupervised toponym linking on two benchmark datasets: Local Global Corpus (LGL; Lieberman et al. 2010) and Wikipedia To-

---

[5]We only consider the place names associated with valid geocoordinates as *toponyms* and do not count the literal ex-pression (e.g., the word "street", "blocks" and "intersection") as toponyms.

[6]We use the unsupervised pretrained weights.

[7]Although SapBERT is trained with biomedical corpus, it generalizes well on geo-entity recognition and linking since the study of diseases often relates to places and regions.

| **GeoWebNews** | Token(B-topo) | | | Token (I-topo) | | | micro- | Entity | | |
|---|---|---|---|---|---|---|---|---|---|---|
| | **Prec** | **Recall** | **F1** | **Prec** | **Recall** | **F1** | **F1** | **Prec** | **Recall** | **F1** |
| BERT | 90.00 | 89.28 | 89.64 | 78.55 | 79.44 | 78.99 | 84.46 | 77.03 | 83.42 | 80.10 |
| SimCSE-BERT | 83.86 | **90.26** | 86.95 | 74.61 | 82.07 | 78.16 | 82.67 | 72.76 | 83.68 | 77.84 |
| SpanBERT | 85.98 | 88.37 | 87.16 | **86.13** | **89.19** | **87.63** | **87.38** | 75.32 | 81.16 | 78.13 |
| SapBERT | 83.12 | 88.32 | 85.64 | 76.26 | 81.11 | 78.61 | 82.22 | 72.48 | 80.16 | 76.12 |
| GEOLM | **91.15** | 90.43 | **90.79** | 79.16 | 84.27 | 81.63 | 86.33 | **82.18** | **85.67** | **83.89** |

Table 1: Toponym recognition results on GeoWebNews dataset. **Bolded** and underlined numbers are for best and second best scores respectively.

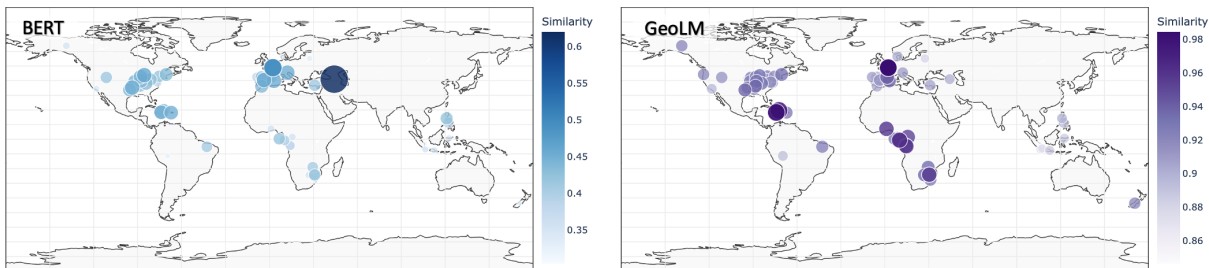

**(Paris, France) Input:** Paris (French: [paʁi]) is the capital and most populous city of France. Situated on the Seine River, in the north of the country, it is in the centre of the Île-de-France region, also known as the région parisienne, "Paris Region"…

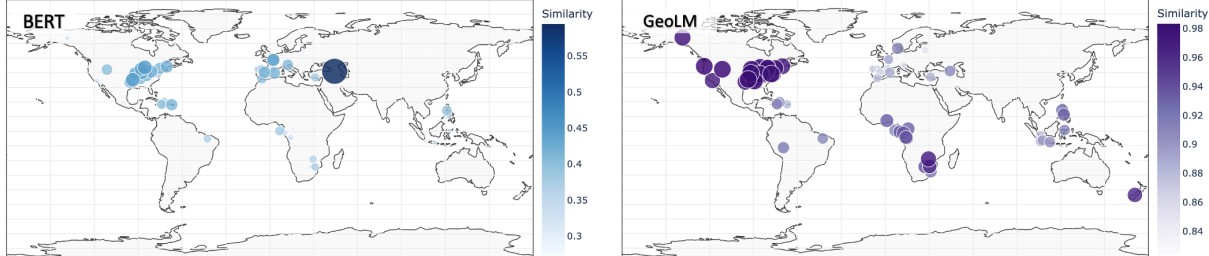

**(Paris, AK, US) Input:** Paris is a city in Logan County, Arkansas, United States, and serves as the county seat for the northern district of Logan County; its southern district counterpart is Booneville…

Figure 4: Visualization of BERT and GEOLM predictions on two "Paris" samples from the WikToR dataset. The two left figures are BERT results, and the two right figures are GEOLM results. The colored circles in the figure denote all the geo-entities named "Paris" in the GeoNames database. The circle's color represents the cosine similarity (or ranking score) between the candidate geo-entity from GeoNames and the query toponym *Paris*. It is worth noting that the BERT predictions are almost the same given the two sentences because BERT does not align the linguistic context with the geospatial context.

ponym Retrieval (WikToR; Gritta et al. 2018b). LGL contains 588 news articles with 4,462 toponyms that have been linked to the GeoNames database. This dataset has ground-truth geocoordinates and GeoNames ID annotations. This dataset has only one sample with a unique name (among 4,462 samples in total). On average, each toponym corresponds to 27.52 geo-entities with the same in GeoNames. WikToR is derived from 5,000 Wikipedia pages, where each page describes a unique place. This dataset also contains many toponyms that share the same lexical form but refer to distinct geo-entities. For example, Santa Maria, Lima, and Paris are the top ambiguous place names. For WikToR, there is no sample with a unique name. The least ambiguous one has four candidate geo-entities with the same name in GeoNames. On

average, each toponym corresponds to 70.45 geo-entities with the same name in GeoNames. This dataset is not linked with the GeoNames database, but the ground-truth geocoordinates of the toponym are provided instead.

**Evaluation Metrics.** For LGL, we evaluate model performance using two metrics. 1) **R@**$k$ is a standard retrieval metric that computes the recall of the top-$k$ prediction based on the ground-truth GeoNames ID. 2) **P@D**, following previous studies (Gritta et al., 2018b), computes the distance-based approximation rate of the top-ranked prediction. If the prediction falls within the distance threshold **D** from the ground truth geocoordinates, we consider the prediction as a correct approximation.[8]

---

[8]Here, **P@D**$_{161}$ is the same as $Acc@161km$ (or $Acc@100miles$ ) reported by Gritta et al. (2018b).

| *LGL* | R@1 | R@5 | R@10 | P@D$_{161}$ |
|---|---|---|---|---|
| BERT | 34.6 | **67.5** | **78.1** | 41.2 |
| RoBERTa | 24.2 | 48.7 | 60.6 | 27.9 |
| SpanBERT | 25.2 | 48.8 | 61.0 | 28.8 |
| SapBERT | 30.8 | 58.8 | 72.2 | 35.1 |
| GEOLM | **38.2** | 65.3 | 72.6 | **44.1** |

| *WikToR* | P@D$_{20}$ | P@D$_{50}$ | P@D$_{100}$ | P@D$_{161}$ |
|---|---|---|---|---|
| BERT | 16.1 | 16.3 | 16.9 | 17.6 |
| RoBERTa | 11.7 | 11.9 | 12.4 | 13.0 |
| SpanBERT | 5.5 | 5.7 | 5.9 | 6.3 |
| SapBERT | 25.9 | 26.3 | 27.0 | 28.3 |
| GEOLM | **32.5** | **33.4** | **34.3** | **35.8** |

Table 2: Toponym linking results on LGL and Wik-TOR datasets. **Bolded** numbers are the best scores and underlined numbers are the second-best scores. R@$k$ measures whether ground-truth GeoNames ID presents among the top $k$ retrieval results. P@D measures whether the top retrieval is within the distance **D** from the ground-truth. $D_{20}$, $D_{50}$, $D_{100}$ and $D_{161}$ indicate the distance thresholds of {20, 50, 100, 161} km.

For WikToR, the ground-truth geocoordinates are given, and we follow Gritta et al. (2020, 2018b) to report **P@D** metrics with various **D** values.

**Models in Comparison.** We compare GEOLM with multiple PLMs, including BERT, RoBETRa, SpanBERT (Joshi et al., 2020) and SapBERT (Liu et al., 2021). In the experiments, we use all base versions of the above models. For a fair comparison, to calculate the representation of the candidate entity, we concatenate the center entity's name and its neighbors' names then input the concatenated sequence (i.e., pseudo-sentence) to all baseline PLMs to provide the linguistic context, following the same unsupervised procedure as in §2.4.

**Results and Discussion.** One challenge of this task is that the input sentences may not have any information about their neighbor entities; thus, instead of only considering the linguistic context or relying on neighboring entities to appear in the sentence, GEOLM's novel approach aligns the linguistic and geospatial context so that GEOLM can effectively map the geo-entity embeddings learned from the linguistic context in articles to the embeddings learned from the geospatial context in geographic data and perform linking. The contrastive learning process during pretraining is designed to help the context alignment. From Tab. 2, GEOLM offers more reliable performance than baselines. On WikTor and LGL datasets, GEOLM obtains the best or second-best scores in all metrics. On LGL, GEOLM demonstrates more precise top-1 retrieval (**R@1**) than other models, and also the highest

scores on **P@D**$_{161}$. Since baselines are able to harness only the linguistic context, while GEOLM can use the linguistic and geospatial context when taking sentences as input. The improvement of **R@1** and **P@D**$_{161}$ from BERT shows the effectiveness of the geospatial context. On WikToR, GEOLM performs the best on all metrics. Since WikToR has many geo-entities with the same names, the scores of GEOLM indicate strong disambiguation capability obtained from aligning the linguistic and geospatial context.

Fig. 4 shows the visualization of the toponym linking results given "Paris" mentioned in two sentences. The input sentences are provided in the figures. Apparently, the first Paris should be linked to Paris, France, and the second Paris goes to Paris, AK, US. However, the BERT model fails to ground these two mentions into the correct locations. BERT ranks the candidate geo-entities only slightly differently for the two input sentences, indicating that BERT relies more on the geo-entity name rather than the linguistic context when performing prediction. On the other hand, our model could predict the correct location of geo-entities. This is because even though the lexical forms of the geo-entity names (i.e., Paris) are the same, the linguistic context describing the geo-entity are distinct. Fig. 4 demonstrate that the contrastive learning helps GEOLM to map the linguistic context to the geospatial context in the embedding space.

### 3.3 Geo-entity Typing

**Task Setup.** We apply GEOLM on the supervised geo-entity typing dataset released by Li et al. (2022b). The goal is to classify the type of the center geo-entity providing the geospatial neighbors as context. We linearize the set of geo-entities to a pseudo-sentence that represent the geospatial context for the center geo-entity then feed the pseudo-sentence into GEOLM. There are 33,598 pseudo-sentences for nine amenity classes, with 80% for training and 20% for testing. We train multiple language models to perform amenity-type classification for the center geo-entity.

**Evaluation Metric.** Following (Li et al., 2022b), we report the F1 score for each class and the micro F1 for all the samples.

**Models in Comparison.** In addition to BERT, SpanBERT and SimCSE-BERT, this task also takes LUKE (Yamada et al., 2020) and SpaBERT (Li

| Classes → | Edu. | Ent. | Fac. | Fin. | Hea. | Pub. | Sus. | Tra. | Was. | Micro F1 |
|---|---|---|---|---|---|---|---|---|---|---|
| BERT | 67.4 | 63.4 | **76.3** | 92.9 | 85.6 | 87.2 | 85.6 | 86.2 | 67.8 | 83.5 |
| SpanBERT | 63.3 | 58.9 | 60.8 | 91.6 | 85.9 | 88.2 | 82.4 | 86.7 | **73.5** | 81.9 |
| SimCSE-BERT | 62.3 | 59.0 | 50.4 | 92.5 | 86.7 | 85.2 | 85.7 | 81.0 | 47.0 | 81.0 |
| LUKE | 64.8 | 60.8 | 59.8 | 94.5 | 85.7 | 86.7 | 85.4 | 85.1 | 51.7 | 82.5 |
| SpaBERT | 67.4 | 65.3 | 68.0 | 95.9 | 86.5 | **90.0** | 88.3 | 88.8 | 70.3 | 85.2 |
| GEOLM | **72.5** | **70.9** | 73.0 | **97.8** | **91.5** | 83.6 | **90.5** | **90.8** | 62.2 | **87.8** |

Table 3: Comparing GEOLM with the state-of-the-art LMs on geo-entity typing. Column names are the OSM classes (education, entertainment, facility, financial, healthcare, public service, sustenance, transportation and waste management). **Bolded** and underlined numbers are for best and second best scores respectively.

| Settings | Prec | Recall | F1 |
|---|---|---|---|
| GEOLM | 82.18 | 85.67 | 83.89 |
| GEOLM (No Contrastive) | 70.67 | 77.98 | 74.14 |
| GEOLM (No Spatial Embed.) | 75.05 | 87.00 | 80.59 |

Table 4: Ablation study on toponym recognition to show the GEOLM performance after removing various components.

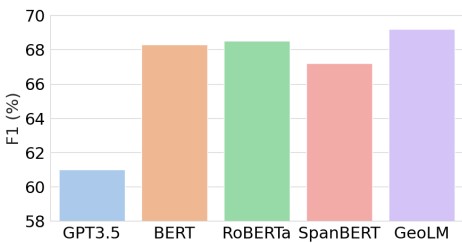

Figure 5: Relation extraction results on SpatialML comparing with other baseline models.

et al., 2022b) into comparison. LUKE is designed to solve entity-related tasks with a specially designed entity tokenizer. SpaBERT generates geo-entity representations given small geographical regions as input. In addition, BERT, SpanBERT, SimCSE-BERT, and LUKE rely only on linguistic information, and SpaBERT relies only on geospatial context to make predictions.

**Results and Discussion.** Tab. 3 shows that the performance of GEOLM surpasses the baseline models using only linguistic or geospatial information. The experiment demonstrates that combining the information from both modalities and aligning the context is useful for geo-entity type inference. Compared to the second best model, SpaBERT, GEOLM has robust improvement on seven types. The result indicates that the contrastive learning during pretraining helps GEOLM to align the linguistic context and geospatial context, and although *only* geospatial context is provided as input during inference, GEOLM can still employ the aligned linguistic context to facilitate the type prediction.

### 3.4 Geospatial Relation Extraction

**Task Setup.** We apply GEOLM to the SpatialML topological relation extraction dataset released by Mani et al. (2010). Given a sentence containing multiple entities, the model classifies the relations between each pair of entities into six types[9] (includ-

ing an NA class indicating that there is no relation). The dataset consists of 1,528 sentences, with 80% for training and 20% for testing. Within the dataset, there are a total of 10,592 entity pairs, of which 10,232 pairs do not exhibit any relation.

**Evaluation Metric.** We report the micro F1 score on the test dataset. We exclude the NA instances when calculating the F1 score.

**Models in Comparison.** We compare GEOLM with other pretrained language models, including BERT, RoBERTa, and SpanBERT, as well as a large language model (LLM), GPT-3.5.

**Results and Discussion.** Fig. 5 shows that GE-OLM achieves the best F1. The results suggest that GEOLM effectively captures the topological relationships between entities. Furthermore, GEOLM demonstrates a better ability to learn geospatial grounding between the mentioned entities.

### 3.5 Ablation Study

We conduct two ablation experiments to validate the model design using toponym recognition and toponym linking: 1) removing the spatial coordinate embedding layer that takes the geo coordinates as input during the training; 2) removing the contrastive loss that encourages the model to learn similar geo-entity embeddings from two types of context (i.e., linguistic context and geospatial context) and only applying MLM on the NL corpora.

For the toponym recognition task, we compare

---

[9]The types come from the RCC8 relations, which contain eight types of geospatial relations. The SpatialML dataset merges four of the relations (TPP, TPPi, NTTP, NTTPi) into the "IN" relation thus resulting in five geo-relations in the dataset.

the entity-level precision, recall, and F1 scores on the GeoWebNews dataset. Tab. 4 shows that removing either component could cause performance degradation. For the toponym linking task, in the first ablation experiment, the linking accuracy (P@D161, i.e., Acc@161km or Acc@100miles) drops from 0.358 to 0.321 after removing the spatial coordinate embedding, indicating that the geocoordinate information is beneficial and our embedding layer design is effective. In the second ablation experiment, the linking accuracy (P@D161) drops from 0.358 to 0.146, showing that contrastive learning is essential.

## 4    Related Work

**Geospatial NLU.** Understanding geospatial concepts in natural language has long been a topic of interest. Previous studies (Liu et al., 2022; Hu, 2018; Wang and Hu, 2019) have used general-purpose NER tools such as Stanford NER (Finkel et al., 2005) and NeuroNER (Dernoncourt et al., 2017) to identify toponyms in text. In the geographic information system (GIS) domain, tools such as the Edinburgh geoparser (Grover et al., 2010; Tobin et al., 2010a), Yahoo! Placemaker[10] and Mordecai (Halterman, 2017, 2023) have been developed to detect toponyms and link them to geographical databases. Also, several heuristics-based approaches have been proposed (Woodruff and Plaunt, 1994; Amitay et al., 2004; Tobin et al., 2010b) to limit the spatial range of gazetteers and associate the toponym with the most likely geo-entity (e.g., most populous ones). More recently, deep learning models have been adopted to establish the connection between extracted toponyms and geographical databases (Gritta et al., 2017, 2018a; DeLozier et al., 2015). For instance, TopoCluster (DeLozier et al., 2015) learns the association between words and geographic locations, deriving a geographic likelihood for each word in the vocabulary. Similarly, CamCoder (Gritta et al., 2018a) introduces an algorithm that encodes toponym mentions in a geodesic vector space, predicting the final location based on geodesic and lexical features. These models utilize supervised training, which assumes that the testing and training data cover the same region. However, this assumption may limit their applicability in scenarios where the testing and training regions differ. Furthermore, Yu

and Lu (2015) use keyword extraction approach and Yang et al. (2022) use language model based (e.g., BERT) classification to solve the geospatial relation extraction problem. However, these models often struggle to incorporate the geospatial context of the text during inference. Our previous work SpaBERT (Li et al., 2022b) is related to GEOLM in terms of representing the geospatial context. It is a language model trained on geographical datasets. Although SpaBERT can learn geospatial information, it does not fully employ linguistic information during inference.

GEOLM is specifically designed to align the linguistic and geospatial context within a joint embedding space through MLM and contrastive learning.

**Language Grounding.** Language grounding involves mapping the NL component to other modalities, and it encompasses several areas of research. In particular, vision-language grounding has gained significant attention, with popular approaches including contrastive learning (Radford et al., 2021; Jia et al., 2021; You et al., 2022; Li et al., 2022a) and (masked) vision-language model on distantly parallel multi-modal corpora (Chen et al., 2020, 2021; Su et al., 2020; Li et al., 2020). Additionally, knowledge graph grounding has been explored with similar strategies (He et al., 2021; Wang et al., 2021). GEOLM leverages both contrastive learning and MLM on distantly parallel geospatial and linguistic data, and it represents a pilot study on the grounded understanding of these two modalities.

## 5    Conclusion

In this paper, we propose GEOLM, a PLM for geospatially grounded language understanding. This model can handle both natural language inputs and geospatial data inputs, and provide connected representation for both modalities. Technically, GEOLM conducts contrastive and MLM pretraining on a massive collection of distantly parallel language and geospatial corpora, and incorporates a geocoordinate embedding mechanism for improved spatial characterization. Through evaluations on four important downstream tasks, toponym recognition, toponym linking, geo-entity typing and geospatial relation extraction, GEOLM has demonstrated competent and comprehensive abilities for geospatially grounded NLU. In the future, we plan to extend the work to question answering and autoregressive tasks.

---

[10]Yahoo! placemaker: https://simonwillison.net/2009/May/20/placemaker/

## Limitations

The current version of our model only uses point geometry to represent the geospatial context, ignoring polygons and polylines. Future work can expand the model's capabilities to handle those complex geometries. Also, it is important to note that the OpenStreetMap data were collected through crowdsourcing, which introduces possible labeling noise and bias. Lastly, model pre-training was conducted on a GPU with at least 24GB of memory. Attempting to train the model on GPUs with smaller memory may lead to memory constraints and degraded performance.

## Ethics Statement

The model weights in our research are initialized from a pretrained BERT model for English. In addition, the training data used in our research are primarily extracted from crowd-sourced databases, including OpenStreetMap (OSM), Wikipedia, and Wikidata. Although these sources provide extensive geographical coverage, the geographical distribution of training data exhibits significant disparities, with Europe having the most abundant data. At the same time, Central America and Antarctica are severely underrepresented, with less than 1% of the number of samples compared to Europe. This uneven training data distribution may introduce biases in the model's performance, particularly in regions with limited annotated samples.

## Acknowledgements

We appreciate the reviewers for their insightful comments and suggestions. We thank the Minnesota Supercomputing Institute (MSI) for providing resources that contributed to the research results reported in this article. Zekun Li and Yao-Yi Chiang were supported by the University of Minnesota Computer Science & Engineering Faculty startup funds. Wenxuan Zhou and Muhao Chen were supported by the NSF Grant IIS 2105329, the NSF Grant ITE 2333736, and the DARPA MCS program under Contract No. N660011924033 with the United States Office of Naval Research, a Cisco Research Award, two Amazon Research Awards, and a Keston Research Award.

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

## A    Distribution of Geo-entities

We analyze the distribution of geo-entities in the pretraining corpora and the downstream datasets. When considering only the name (without geocoordinates), the overlapping percentages of the pretraining data and downstream datasets are shown in Tab. 5. Ablation experiments show that for the geoentities already included during the pretraining, the average P@D161 score is 0.362. For the ones that are not included during the pretraining, the average P@D161 score is 0.341, which is not significantly different from the prior one. This indicates that the improved performance of GEOLM comparing with other models benefits from enhancing the geospatial representations.

## B    Comparison with Other Methods

For the toponym recognition task, we compare the results of GEOLM with other existing models designed specifically for this problem. According to (Gritta et al., 2020), the token-level F1 achieved by Yahoo! Placemaker, Edinburgh Geoparser, Spacy NLP, Google Cloud Natural Language, and NCRF++ are 63.2%, 63.6%, 74.9%, 83.2% and 88.6% respectively. GEOLM has a token-level F1 of 86.3%, which is better than all existing ones except NCRF++. The reason is that NCRF++ uses fine-grained toponym taxonomy to boost the toponym recognition performance. However, the finegrained labels can sometimes be difficult to collect for large-scale datasets. In addition, NCRF++ is a specific toponym recognition model that does not support other geospatial-inference tasks. With GEOLM, we can generate representations useful for various tasks.

For the toponym linking task, we compare our model with the other existing geoparser models mentioned in EUPEG (Hu, 2018) and Voting (Hu et al., 2023), including CLAVIN[11], TopoCluster (DeLozier et al., 2015), CamCoder (Gritta et al., 2018a), Modecai (Halterman, 2017), GENRE (De Cao et al., 2020), and Voting (Hu et al., 2023). Since EUPEG evaluates the toponym resolution and toponym linking together and does not provide

---

[11]CLAVIN:https://github.com/Novetta/CLAVIN

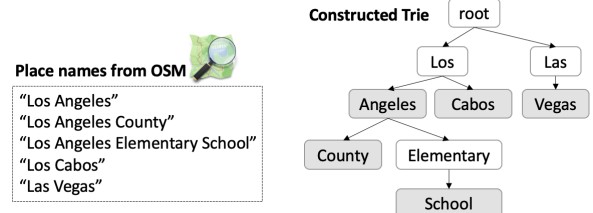

Figure 6: Example of the constructed Trie if OpenStreetMap only contains "Los Angeles", "Los Angeles County", "Los Angeles Elementary School", "Los Cabos" and "Las Vegas". Each node is a word in a place name. The gray color indicates that this node is the last word of a place name.

the scores for linking only, we use the scores reported in Voting, which assumes gold toponyms as inputs for toponym linking (same as ours).

The scores in P@D161 (or Accuracy@161km) are shown in Tab. 6. The models that perform better than GEOLM are all supervised learning models (i.e. CamCoder, GENRE and Voting) while GEOLM is unsupervised. Within the unsupervised group, GEOLM performs the best. The benefit of the unsupervised nature of GEOLM is that GEOLM can handle new samples without the need for extra training data, which is often difficult to gather. Also, new geo-entity names can appear in documents and the way people call the same geo-entity can change over time, the unsupervised approach has the advantage of handling ever-changing documents, e.g., online text, while the supervised approach focuses on existing names of geo-entities. We acknowledge that there is a gap between GEOLM and the domain-specific geoparsers, and we will aim to narrow this gap in the future.

## C    Trie

Trie supports a tree structure for efficient search for geo-entity names, and it helps distinguish between two geo-entities with shared substrings in their names. For example, Trie helps extract the geo-entity "Los Angeles High School" from the sentence "I work at the Los Angeles High School," instead of extracting the geo-entity "Los Angeles". Fig. 6 shows an example Trie constructed from a set of geo-entity names.

We use all geo-entity names in the worldwide OpenStreetMap database to construct a worldwide Trie, where each node is a single word in the name. When using Trie to preprocess the Wikipedia documents, we apply the Trie searching to find all

| Tasks | Dataset | # Total Records | # Intersection | Percentage |
|---|---|---|---|---|
| Toponym Recognition | GeoWebNews | 912 | 325 | 35.6% |
| Toponym Linking | WikToR | 1886 | 1280 | 67.9% |
| Geo-entity Typing | OpenStreetMap Subset | 23195 | 544 | 2.3% |
| Relation Extraction | SpatialML | 309 | 91 | 29.4% |

Table 5: The number of geo-entities in the downstream datasets, and the overlapping portion of geo-entities with the pretraining corpora.

| Unsupervised | | Supervised | |
|---|---|---|---|
| Modecai | 0.15 | CamCoder | 0.63 |
| CLAVIN | 0.22 | GENRE | 0.81 |
| TopoCluster | 0.24 | Voting | 0.85 |
| GEOLM (ours) | 0.35 | | |

Table 6: Comparison with other toponym linking models in both unsupervised and supervised category.

| Types | Keywords |
|---|---|
| Others | Other; No relation |
| Equal (EQ) | EQ |
| Disconnected (DC) | DC |
| Externally connected (EC) | EC |
| Within (IN) | In; Within |
| Partially overlapping (PO) | PO |

Table 7: Relation types and the corresponding acceptable keywords in GPT outputs

the mentioned geo-entity names. To mitigate the disambiguation error, we use the Wikipedia page title to filter out the mentions that do not describe the "entity-of-interest". After this step, the disambiguation error only occurs when two distinct geo-entities with the same name occur on the same Wikipedia page, which is pretty rare. This does not fully resolve the disambiguation error issue, and there may still be noises in the training data. However, as long as most of the data is clean, the model can still learn meaningful information from the data. Thus we have compiled a quite large pretraining NL corpora with 1,458,150 sentences/paragraphs describing 472,067 geo-entities.

## D GPT-3.5 Prompt

We use GPT-3.5-turbo to help predict the relations between two geo-entities. With the *system* role, we prompt the model with the general task description and ask the model to choose from one of the possible relationships. With the *user* role, we provide the input sentence and the two geo-entity names. The example prompt is shown below.

- **System**: *Given a sentence, and two entities within the sentence, classify the relationship between the two entities based on the provided sentence. All possible Relationships are*

listed below: [ disconnected (DC): Entity A and B have no spatial intersection, both in terms of interiors and boundaries; externally connected (EC): Entity A and B touch each other only at their boundaries; equal (EQ): Entity A and B are identical; partially overlapping (PO): Interiors of entity A and B overlap but neither is completely contained within the other; within (IN): One entity is part of the other entity; Others: No relation between entities]

- **User**: *Sentence: {input-sentence} Entity1: {entity1-name} Entity2: {entity2-name} Relationship:*

To address the randomness in the GPT outputs and robustly evaluate the performance, we only look for some particular keywords from the GPT outputs. If the output contains the desired keyword, we consider the prediction as correct. Tab. 7 lists the keywords for each relation type.