# OpenReview forum: "GeoLM: Empowering Language Models for Geospatially Grounded Language Understanding"
_EMNLP/2023/Conference — EMNLP 2023 Main_

### Official Review · Reviewer_TG2x · 2023-07-20

**Soundness:** 3

**Excitement:**

3: Ambivalent: It has merits (e.g., it reports state-of-the-art results, the idea is nice), but there are key weaknesses (e.g., it describes incremental work), and it can significantly benefit from another round of revision. However, I won't object to accepting it if my co-reviewers champion it.

**Paper Topic And Main Contributions:**

The paper proposes a new language model that is trained on text data that includes geo-entities. Specifically, the approach merges data from Wikipedia, Wikidata, and Open Street Map where every data point is represented as text data. The proposed GeoLM model is pre-trained on this collected data using BERT model as a base. The training procedure includes two tasks: contrastive learning and masked language modelling. The approach is evaluated on 4 down-stream tasks that are related to parsing geo-entities in text.

The main contribution is the proposed GeoLM language model that is pre-trained on the collected data points.

**Questions For The Authors:**

Are the compared models in section 3.1, 3.3 and 3.4 also fine-tuned on respective down-stream tasks?

**Reasons To Accept:**

The paper is well-written with clear definitions and details for the proposed approach as well as the experimental evaluation on four tasks. It is also shown that the model achieves somewhat improved performance with compared counterparts.

**Reasons To Reject:**

The main criticism would about the compared models: BERT, SpanBERT, RoBERTa, SapBERT, LUKE. The Section 3 provides details about each down-stream task and the acquired results presented in tables. It is not really clear whether the compared models were also fine-tuned on the respective task or not. For example, the Section 3.1 clearly states that GeoLM is trained on 80% and tested on 20%. But it is not clear whether the compared models were also trained on the down-stream task of toponym recognition. The same detail is missing for 3.3 and 3.4. Section 3.2 clearly mentions that it is performed in an unsupervised fashion.

**Reproducibility:**

5: Could easily reproduce the results.

**Reviewer Confidence:**

5: Positive that my evaluation is correct. I read the paper very carefully and I am very familiar with related work.

---

> ### Author Rebuttal · Authors · 2023-08-29
>
> **Q: Are the compared models in section 3.1, 3.3 and 3.4 also fine-tuned on respective down-stream tasks?**
>
> Yes, all the compared models in Section 3.1, 3.3, and 3.4 are fine-tuned on the same task training data as GeoLM and GeoLM outperforms these compared models. We’ll make that clear in the revised version.

---

### Official Review · Reviewer_HVHc · 2023-08-04

**Soundness:** 4

**Excitement:**

3: Ambivalent: It has merits (e.g., it reports state-of-the-art results, the idea is nice), but there are key weaknesses (e.g., it describes incremental work), and it can significantly benefit from another round of revision. However, I won't object to accepting it if my co-reviewers champion it.

**Missing References:**

There's a recent paper that discusses current challenges associated to text geoparsing, which should be analyzed in the context of this work:

Geoparsing: Solved or Biased? An Evaluation of Geographic Biases in Geoparsing
https://agile-giss.copernicus.org/articles/3/9/2022/

There's also paper describing a benchmark platform named EUPEG for evaluating text geoparsing over multiple datasets, going beyond the datasets used in the present paper. Ideally, the authors should compare their results on toponym recognition/disambiguation against results reported for other systems/approaches, and EUPEG can perhaps help on this:

Enhancing spatial and textual analysis with EUPEG: an extensible and unified platform for evaluating geoparsers
https://geoai.geog.buffalo.edu/EUPEG/
https://github.com/geoai-lab/EUPEG

Another recent system for toponym recognition and disambiguation is Mordecai, which is conveniently available as open-source software. Passing texts through Mordecai and comparing results against the proposed approach would also be relatively simple:

Mordecai 3: A Neural Geoparser and Event Geocoder
https://arxiv.org/pdf/2303.13675.pdf
https://github.com/ahalterman/mordecai3

**Paper Topic And Main Contributions:**

The paper presents GEOLM, a Transformer encoder model that is pre-trained with language descriptions of geospatial locations, paired with geospatial information extracted from geographical databases such as OSM or Wikidata, in an attempt to ground the language model and enhance the understanding of geospatial entities in natural language. The pre-training procedure leverages masked language modeling together with a contrastive objective, where ??. The evaluation of the model considered different tasks (i.e., toponym recognition, toponym disambiguation/linking, geospatial relation extraction, and geo-entity typing), and results show that the proposed model outperforms standard BERT and other extensions focused on better modeling entities and/or spans of text.

**Questions For The Authors:**

A - In Section 2.2, the authors mention the use of "trie-based phrase matching" to find the sentences that contain the name of the target geo-entity. This procedure should ideally be explained in more detail, and the authors should comment on the fact that this procedure can also produce disambiguation errors.

B - Section 2.3 should explain the combination of the two pre-training tasks in more detail. The contrastive learning objective seems to use in batch negatives, and the linguistic/verbalized sentences need thus to be encoded separately by the model. On the other hand, the MLM objective uses the concatenation of the linguistic/verbalized sentences and they are encoder jointly by the model. How are then the two tasks combined together when processing a batch of instances? Also, the use of negative samples should be explained in more detail (e.g., how are the hard negative samples selected).

C - In Section 2.4, the procedure associated to toponym linking should also be explained in more detail. How are the candidate entities from geographical data encoded by GEOLM (e.g., does the model encode simply the name for the entity? Or instead the name together with a textual description? Or perhaps a sentence verbalizing the entity name together with names for neighbouring entities, as done during pre-training?)

D - Also in Section 2.4, the relation extraction task should be better explained. What does it mean to "concatenate the average embedding of entities in the final layer of GEOLM"?

E - The toponym recognition and toponym linking tasks, in particular, should ideally be evaluated with more datasets. Results should also be compared against those from previous methods specifically focusing on these tasks, instead of just making the comparison against BERT/SpanBERT and the previous work named SapBERT. Results against SapBERT are encouraging, but ideally the paper should also explain if the baselines were fine-tuned to the in-domain data in the same way as GEOLM.

F - The experimental evaluation should ideally consider an ablation in which GEOLM is just pre-trained with masked language modeling on the same Wikipedia data, instead of considering the contrastive objective and the verbalized sentence. A separate ablation should ideally also consider an alternative which does not use the geographic/coordinate embeddings.

**Reasons To Accept:**

* The paper is well written, mostly clear in its descriptions, and it addresses an interesting problem.

* The experimental results are encouraging. The paper assesses the model on many different tasks, with positive results in comparison to a previous related work named SPABERT (Li et al., 2022b).

**Reasons To Reject:**

* The experimental evaluation in the toponym recognition and disambiguation tasks is not particularly convincing, since the authors do not compare results against previous studies specifically focusing on these problems (e.g., see previous benchmarks such as EUPEG).

* The descriptions provided in the paper are not entirely convincing, in terms of justifying why the use of the verbalized sentences and/or the coordinate embeddings provides any benefits. Ablation experiments focusing on these components would help in assessing this claim.

**Reproducibility:**

3: Could reproduce the results with some difficulty. The settings of parameters are underspecified or subjectively determined; the training/evaluation data are not widely available.

**Reviewer Confidence:**

4: Quite sure. I tried to check the important points carefully. It's unlikely, though conceivable, that I missed something that should affect my ratings.

**Typos Grammar Style And Presentation Improvements:**

The following comment is not exactly a typo, but instead a suggestion in terms of how the motivation is framed in the introduction of the paper.

Specifically, the authors state that some geo-corpora are available (e.g., LGL and SpatialML), but they only cover limited geographical regions. I would argue that this is not the main limitation, given that some of the existing corpora are of global coverage. The main issue is, nonetheless, the fact that the existing geo-corpora are relatively small. Also in the introduction the authors state that Wikipedia does not always contain geo-coordinates and spatial relations between geo-entities, but again this is not the main limitation (i.e., many entities in Wikipedia, not just places, are indeed associated to geospatial coordinates of latitude and longitude). The issue with Wikipedia is that its descriptions constitute a very specific genre of text, different from other documents where one may which to use geographic text analysis methods.

---

> ### Author Rebuttal · Authors · 2023-08-29
>
> **Q1: the toponym recognition and disambiguation tasks did not compare with results against previous studies specifically focusing on these problems (e.g., see previous benchmarks such as EUPEG).**
>
> For the **toponym recognition task**, we compare the results of GeoLM with other existing models designed specifically for this problem. According to Gritta et al. [1], the token-level F1 achieved by Yahoo! Placemaker, Edinburgh Geoparser, Spacy NLP, Google Cloud Natural Language, and NCRF++ are 63.2%, 63.6%, 74.9%, 83.2% and 88.6% respectively. GeoLM has a token-level F1 of 86.3%, which is better than all existing ones except NCRF++. The reason is that NCRF++ uses fine-grained toponym taxonomy to boost the toponym recognition performance. However, the fine-grained labels can sometimes be difficult to collect for large-scale datasets. In addition, NCRF++ is a toponym recognition specific model which does not support other geospatial-inference tasks. With GeoLM, we can generate representations useful for various tasks.
>
>
> For the **disambiguation (toponym linking)** task, we compare our model with the other existing geoparser models mentioned in EUPEG [2] and Voting [3], including CLAVIN, TopoCluster, CamCoder, Modecai,  GENRE, and Voting. Since EUPEG evaluates the toponym resolution and toponym linking together and does not provide the scores for linking only, we use the scores reported in Voting, which assumes gold toponyms as inputs for toponym linking (same as ours).
>
> The scores in P@D161 (or Accuracy@161km) are shown in the table below:
>
> | Unsupervised | | | |
> | -------- | ------- | ------- |------- |
> | GeoLM (ours)     | CLAVIN | TopoCluster | Modecai |
> | **0.35** | 0.22 | 0.24 | 0.15 |
>
> | Supervised | | |
> |------- |------- | ------- |
> | CamCoder | GENRE | Voting |
> | 0.63 | 0.81  | 0.85 |
>
>
> Note that the models perform better than GeoLM are all supervised learning models (i.e. CamCoder, GENRE and Voting) while GeoLM is unsupervised. Within the unsupervised group, GeoLM performs the best. The benefit of the unsupervised nature of GeoLM is that it can handle new samples without the need of extra training data, which is often difficult to gather. Also, *new* geo-entity names can appear in documents and the way people call the same geo-entity can *change* over time, the unsupervised approach has the advantage of handling ever changing documents, e.g., online text, while the supervised approach focuses on existing names of geo-entities. We acknowledge that there is a gap between GeoLM and the domain-specific geoparsers, and we will aim to narrow this gap in the future.
>
> [1] Gritta, Milan, Mohammad Taher Pilehvar, and Nigel Collier. "A pragmatic guide to geoparsing evaluation: Toponyms, Named Entity Recognition and pragmatics." Language resources and evaluation 54 (2020): 683-712.
>
> [2] Wang, Jimin, and Yingjie Hu. "Enhancing spatial and textual analysis with EUPEG: An extensible and unified platform for evaluating geoparsers." Transactions in GIS 23, no. 6 (2019): 1393-1419.
>
> [3] Hu, Xuke, Yeran Sun, Jens Kersten, Zhiyong Zhou, Friederike Klan, and Hongchao Fan. "How can voting mechanisms improve the robustness and generalizability of toponym disambiguation?." International Journal of Applied Earth Observation and Geoinformation 117 (2023): 103191.
>
>
> &nbsp;
>
> **Q2: Justify why the use of the verbalized sentences and/or the coordinate embeddings provides any benefits. The experimental evaluation should ideally consider an ablation in which GEOLM is just pre-trained with masked language modeling on the same Wikipedia data, instead of considering the contrastive objective and the verbalized sentence. A separate ablation should ideally also consider an alternative which does not use the geographic/coordinate embeddings.**
>
>
> A verbalized sentence (pseudo-sentence) is a simple form to linearize a small geospatial region. The order of the place names in the pseudo sentence reflects the distance relation with the pivot entity. The spatial coordinate embedding helps learn the distance and orientation relations between geo-entities.
>
> We add the two ablation experiments as suggested by the reviewer and evaluate the model on the toponym recognition and toponym linking task. In the first experiment, we pretrain the model using only the MLM objective on the NL corpora consisting of Wikipedia and Wikidata. In the second experiment, we remove the spatial coordinate embedding during pretraining and evaluate the model on the downstream tasks.
>
>
> For the **toponym recognition** task,  we compare the entity-level precision, recall, and F1 scores on the GeoWebNews dataset and observe that removing either of the components could cause performance degradation.
>
> | Setting     | Precision |  Recall | F1 |
> | -------- | ------- | ------- | ------- |
> | GeoLM | 82.18 | 85.67 | 83.89 |
> |GeoLM (No Contrastive, only MLM) | 70.67 | 77.98| 74.14 |
> | GeoLM (No Spatial Embedding)| 75.05 | 87.00 | 80.59 |
>
>
> For the **toponym linking** task, in the first ablation experiment, P@D161 dropped from 0.358 to 0.321 after removing the spatial coordinate embedding, indicating that the geocoordinate information is important and our embedding layer design is effective. In the second ablation experiment, P@D161 dropped from 0.358 to 0.146, showing that contrastive learning is essential.
>
> &nbsp;
>
> **Q3: Explain Trie in more detail, comment on the fact that this procedure can also produce disambiguation errors.**
>
> Using Tries has two advantages. First, Tries support a tree structure for efficient search for geo-entity names. Second, Tries helps differentiate between two geo-entities with shared substrings in their names. For example, Tries helps extract the geoentity “Los Angeles High School” from the sentence “I work at the Los Angeles High School,” instead of extracting the geoentity “Los Angeles”.
>
> We use all geo-entity names in the worldwide OpenStreetMap database to construct a worldwide Trie, where each node is a single word in the name. When using Trie to preprocess the Wikipedia documents, we apply the Trie searching to find all the mentioned geo-entity names. To mitigate the disambiguation error, we use the Wikipedia page title to filter out the mentions that do not describe the “entity-of-interest”. After this step, the disambiguation error only occurs when two different geo-entities with the same name occur in the same Wikipedia page, which is pretty rare. We admit that this does not fully resolve the disambiguation error issue, and it may still cause noise in the training data. However, as long as most of the data is clean, the model can still learn meaningful information from the data. Thus we have compiled a quite large pretraining NL corpora with 1,458,150 sentences/paragraphs describing 472,067 geo-entities.
>
> &nbsp;
>
> **Q4: Section 2.3 should explain the combination of the two pre-training tasks in more detail, also explain how in-batch hard negatives are selected**
>
> The two losses are calculated on the same batch of data. With a batch size of 32, we will have 16 samples of NL (linguistic) from Wikipedia & Wikidata and 16 samples of pseudo (verbalized) sentences constructed from OpenStreetMaps (OSM). These samples in two modalities are paired and each pair describes the same geo-entity; thus we have 16 positive pairs in total. Moreover, among the 16 pairs, 8 of them have different geo-entity names (easy negatives), and 8 of them have the same geo-entity name but different OSM ID (hard negatives). To calculate the MLM loss, we merge the NL sentence and pseudo sentence to feed into the model. To calculate the contrastive loss, we keep the NL sentence and pseudo sentence separate and sample from positive and negative pairs to calculate the InfoNCE (contrastive) loss.
>
> &nbsp;
>
> **Q5: How are the candidate entities from geographical data encoded by GEOLM**
>
> Following the convention in SpaBERT [1], to capture the spatial context of candidate entities from a geographical database (e.g., GeoNames) (which does not contain textual descriptions), the model encodes the name for the entity, using the neighbor entity names as the context,  as done during pre-training. The pivot entities are concatenated with the neighbor geo-entities to provide the token input, and their geo-coordinates are embedded with the spatial coordinate embedding layer.
>
> Different from SpaBERT, GeoLM is a novel model that combines both linguistic and spatial context in a contrastive way so that even when the candidate entities are only encoded with spatial context, GeoLM can still match the corresponding entity encoded with linguistic context.
>
> [1] Li, Zekun, Jina Kim, Yao-Yi Chiang, and Muhao Chen. "SpaBERT: A Pretrained Language Model from Geographic Data for Geo-Entity Representation." In Findings of the Association for Computational Linguistics: EMNLP 2022, pages 2757–2769.
>
> &nbsp;
>
> **Q6: Also in Section 2.4, the relation extraction task should be better explained. What does it mean to "concatenate the average embedding of entities in the final layer of GEOLM"?**
>
> We compute the average embedding of tokens to form the entity embedding and then concatenate the embeddings of the subject and object entities to use as input for the classifier. We'll make it clear in the revision.
>
> &nbsp;
>
> **Q7: Missing References**
> We will add the references in the revised version.
>
> &nbsp;
>
> **Q8: Suggestion on how motivation should be framed**
>
> We appreciate this suggestion and agree with the reviewer’s comment. To tackle the problem of small geo-corpora, GeoLM incorporates Wikipedia to constitute the pretraining corpora. As the reviewer pointed out, Wikipedia constitutes a very specific text genre when describing the geo-entity (e.g., history, environment, etc.). GeoLM overcomes this problem by capturing broader relations between geo-entities learned from Wikidata relations and the OpenStreetMap database. Granted that GeoLM does not entirely solve these two challenges. Still, the experiments show that GeoLM provides a pilot solution to these problems, demonstrated in the downstream tasks, including geo-entity linking and relation extraction. We will modify the motivation section of the paper in the revised version.

---

### Official Review · Reviewer_zGPk · 2023-08-05

**Typos Grammar Style And Presentation Improvements:** N/A
**Soundness:** 3

**Excitement:**

3: Ambivalent: It has merits (e.g., it reports state-of-the-art results, the idea is nice), but there are key weaknesses (e.g., it describes incremental work), and it can significantly benefit from another round of revision. However, I won't object to accepting it if my co-reviewers champion it.

**Missing References:**

N/A

**Paper Topic And Main Contributions:**

The paper proposes a geospatially grounded language model for geospatial language understanding named GeoLM. And the authors showed that this pre-trained model can address various geographical-related language understanding tasks by utilizing both geographical and natural language knowledge. GeoLM is tested on various traditional downstream tasks and is shown to be more effective than the compared baselines.

**Questions For The Authors:**

A. What is the distribution of geo-entities in both the pretraining corpora and downstream task datasets and what are the overlapping sets of geo-entities for both data types? Does the improved performance result from introducing more geo-entity knowledge or from enhancing geospatial representations?

B. For the downstream task of toponym linking, could you elaborate on the methods used to calculate representations for all candidate entities across different PLMs? Are all models purely name-based?

C. For spatial relation extraction, it seems that a significant portion of the dataset belongs to the NA class. Is the F1 score shown in Figure 5 calculated excluding the NA class? If the NA class is included, could you provide the score for each individual spatial relation?

D. If during the pre-training process, people can already access the coordinates to the location entities, then for some tasks like relation extraction, why cannot directly get the results from coordinates/maps?

E. For the relation extraction task, in line 482, the author stated that there are 7 types for the relations, but the dataset they mentioned (Mani et al., 2010) only has 5 link types (excluding the NA class). Can the authors list all the link types they have in their experiments?

**Reasons To Accept:**

1. The paper is well-written and the motivation is clear. Geospatial language understanding is an essential task in the geospatial community.
2. The method is shown to be effective and outperforms some previous SOTAs like SpaBERT.
3. The idea of incorporating coordinates in the pre-training is well-formulated.

**Reasons To Reject:**

1. The authors only conducted experiments on the downstream tasks. There are no ablation studies discussing the effectiveness of each module in the paper.
2. See the questions below.

**Reproducibility:**

3: Could reproduce the results with some difficulty. The settings of parameters are underspecified or subjectively determined; the training/evaluation data are not widely available.

**Reviewer Confidence:**

4: Quite sure. I tried to check the important points carefully. It's unlikely, though conceivable, that I missed something that should affect my ratings.

---

> ### Author Rebuttal · Authors · 2023-08-29
>
> **Q1: No ablation studies discuss each module's effectiveness in the paper.**
>
> We conducted two additional ablation studies to show the validity of the model design using the **toponym recognition** and **toponym linking** as the testing ground:
>
> * 1) Removing the spatial coordinate embedding layer that takes the geo coordinates as input during the training
> * 2) Removing the contrastive loss that encourages the model to learn similar geo-entity embeddings from two different types of context (i.e., linguistic context and spatial context) and only applying MLM on the NL corpora
>
> For the toponym recognition task,  we compare the entity-level precision, recall, and F1 scores on the GeoWebNews dataset and observe that removing either of the components could cause performance degradation.
>
>
> | Setting     | Precision |  Recall | F1 |
> | -------- | ------- | ------- | ------- |
> | GeoLM | 82.18 | 85.67 | 83.89 |
> |GeoLM (No Contrastive) | 70.67 | 77.98| 74.14 |
> | GeoLM (No Spatial Embedding)| 75.05 | 87.00 | 80.59 |
>
>
> For the toponym linking task, in the first ablation experiment, the linking accuracy (P@D161, i.e, Acc@161km or Acc@100miles) dropped from 0.358 to 0.321 after removing the spatial coordinate embedding, indicating that the geocoordinate information is beneficial and our embedding layer design is effective.  In the second ablation experiment, the linking accuracy (P@D161) dropped from 0.358 to 0.146, showing that contrastive learning is essential.
>
>
> &nbsp;
>
> **Q2. The distribution of geo-entities in the pretraining corpora and downstream task datasets and the overlapping sets of geo-entities for both data types.**
>
> We analyze the distribution of geo-entities in the pretraining corpora and the downstream tasks. When considering only the name (without geo-coordinates), the overlapping percentages of the pretraining data and downstream datasets are shown in the table below
>
> | Task     | Dataset | # Total Records | # Intersection | Percentage |
> | -------- | ------- | ------- |------- | ------- |
> | Toponym Recognition  | GeoWebNews | 912  | 325  | 35.6%  |
> | Toponym Linking | WikToR     | 1886 | 1280  |67.9%  |
> | Geo-entity Typing    | OpenStreetMap Subset   | 23195 | 544  | 2.3%  |
> | Relation Extraction    | SpatialML   | 309 | 91  | 29.4%  |
>
> We further analyze the toponym linking experiment results. The results show that for the geo-entities already included during the pretraining, the average P@D161 score is 0.362. For the ones that are *not* included during the pretraining, the average P@D161 score is 0.341, which is not significantly different from the prior one. This indicates that the improved performance of GeoLM benefits more from enhancing the geospatial representations.
>
> Regarding the overlapping sets of geo-entities for both data types (i.e., NL corpora and geographical dataset), all the geo-entities exist in both data types to train the model with the contrastive objective.
>
>
> We will add the distribution and analysis in the revised version.
>
> &nbsp;
>
>
> **Q3. For the downstream task of toponym linking, could you elaborate on the methods used to calculate representations for all candidate entities across different PLMs? Are all models purely name-based?**
>
> All of the compared PLMs are purely name-based. For a fair comparison, to calculate the representation of the candidate entity, we concatenate the pivot entity’s name and its neighbors’ names and input the concatenated sequence (i.e., pseudo sentence in the paper) to the model to provide linguistic context, following the same procedure as GeoLM. The positional encoding used here reflects the spatial order of the neighbors to the pivot entity.
>
> &nbsp;
>
> **Q4. Is the F1 score shown in Figure 5 calculated excluding the NA class?**
>
> The F1 score excludes the NA class instances that are treated as negative instances. This is consistent with the common practice of evaluating relation extraction tasks where the majority of entity pairs are not related [1].
>
> [1] Zhang, Yuhao, Victor Zhong, Danqi Chen, Gabor Angeli, and Christopher D. Manning. "Position-aware attention and supervised data improve slot filling." In Conference on Empirical Methods in Natural Language Processing. 2017.
>
> &nbsp;
>
> **Q5. If during the pre-training process, people can already access the coordinates to the location entities, then for some tasks like relation extraction, why cannot directly get the results from coordinates/maps?**
>
> Directly obtaining the results from map data (location entities with coordinates) only works when individual toponyms in the documents are all correctly resolved (i.e., linked to their correct geocoordinates). However, toponym linking (resolution)  is another challenging problem, as we show in Section 3.2. More importantly, documents can contain much richer semantic information about entity relations than map data. For example, “I went to the Disneyland in Los Angeles” is commonly said, reflecting the “IN” relation between these two geo-entities. This represents people’s conceived space (i.e., what is thought) that is useful but can be inconsistent with the perceived space (i.e., what is seen) in map data. In this case, Disneyland is actually located in Orange County, but not in Los Angeles.
>
> &nbsp;
>
> **Q6. For the relation extraction task, in line 482, the author stated that there are 7 types for the relations, but the dataset they mentioned (Mani et al., 2010) only has 5 link types (excluding the NA class). Can the authors list all the link types they have in their experiments?**
>
> We apologize for the typo. There should be six types in total: five geo-relations plus “no-relation”. The types come from RCC8 relations, which contain eight types of geo-spatial relations, and the SpatialML dataset merged four of the relations (TPP, TPPi, NTTP, NTTPi) into “IN” relation thus resulting in five geo-relations in the dataset.

---

### Meta-Review · Area_Chair_Fdfg · 2023-09-21

**Recommendation:** 4

**Metareview:**

This paper presents a new language model (GeoLM) that is pre-trained using masked language modelling and a contrastive objective on descriptions of geospatial locations and information with the goal of enhancing geospatial abilities of the model. This is evaluated using a suite of downstream tasks including toponym recognition, toponym disambiguation, geospatial relation extraction and geo-entity tagging and shows improvements over baselines and existing methods.

All reviewers agree on multiple aspects, including that the paper is well written and motivated, that it addresses a relevant problem and highlight that the results are good in comparison with relevant baselines. Evaluating on four downstream tasks gives robustness to the results.

There were some questions about some details of the work (e.g. trie based matching, evaluation setup) that were addressed well in the author response. Ablation experiments were also mentioned by two of the reviewers, and the response provided results on this aspect as well.  This leads to all reviewers agreeing that the paper methodology and results are sound.

Overall, the authors should consider including the details provided in the discussion in their camera ready version of the work. Beyond this, there are few pending comments from the reviewers.

---

### Decision · Program_Chairs · 2023-10-07

**Decision:**

Accept-Main

**Comment:**

This paper presents a new language model (GeoLM) that is pre-trained using masked language modelling and a contrastive objective on descriptions of geospatial locations and information with the goal of enhancing geospatial abilities of the model. This is evaluated using a suite of downstream tasks including toponym recognition, toponym disambiguation, geospatial relation extraction and geo-entity tagging and shows improvements over baselines and existing methods.

All reviewers agree on multiple aspects, including that the paper is well written and motivated, that it addresses a relevant problem and highlight that the results are good in comparison with relevant baselines. Evaluating on four downstream tasks gives robustness to the results.

There were some questions about some details of the work (e.g. trie based matching, evaluation setup) that were addressed well in the author response. Ablation experiments were also mentioned by two of the reviewers, and the response provided results on this aspect as well.  This leads to all reviewers agreeing that the paper methodology and results are sound.

Overall, the authors should consider including the details provided in the discussion in their camera ready version of the work. Beyond this, there are few pending comments from the reviewers.